Journal of Data-centric Machine Learning Research (2026)          Submitted 07/24; Revised 01/26; Published 03/26

,

# IntelliGraphs: Datasets for Benchmarking Knowledge Graph Generation

**Thiviyan Thanapalasingam**                                    THIVIYAN.T@GMAIL.COM
*University of Amsterdam*

**Emile van Krieken**
*Vrije Universiteit Amsterdam*

**Peter Bloem**
*Vrije Universiteit Amsterdam*

**Paul Groth**
*University of Amsterdam*

**Reviewed on OpenReview:** *https://openreview.net/forum?id=4wbayMJWwN*

**Editor:** Mykola Pechenizkiy

## Abstract

Knowledge Graph Embedding (KGE) models are used to learn continuous representations of entities and relations, commonly trained to predict missing links between entities. However, Knowledge Graphs are not just sets of links but also have complex semantics underlying their structure. Semantics plays a crucial role in several downstream tasks, such as query answering and reasoning. Recognizing this, our work goes beyond simple link prediction to focus on inferred knowledge that adheres to rich semantics. Specifically, 1) we introduce the *subgraph inference* task, where a model is required to generate novel subgraphs that are logically consistent with background knowledge; 2) we propose *IntelliGraphs*, a set of five new datasets that contain subgraphs with logical rules that express complex semantics for evaluating subgraph inference models, and 3) we design four baseline models, which include three models based on traditional KGEs, and show empirically that the KGE-based baselines cannot capture complex semantics. We believe that IntelliGraphs will encourage the development of machine learning models that focus on semantic understanding.

**Keywords:** Knowledge Graph, Logical Constraints, Subgraph Inference, Semantic evaluation

## 1 Introduction

Knowledge Graphs (KGs) contain knowledge about the world structured as graphs with entities connected through different relations (Hogan et al., 2021). Large-scale KGs are widely used in a range of applications, such as query answering (Arakelyan et al., 2020) and information retrieval (Noy et al., 2019).

Knowledge Graph Embedding (KGE) models were proposed to address the problem of incompleteness in KGs. KGE models learn continuous representations for entities and relations (Bordes et al., 2013; Yang et al., 2014) through *link prediction*, the task of predicting

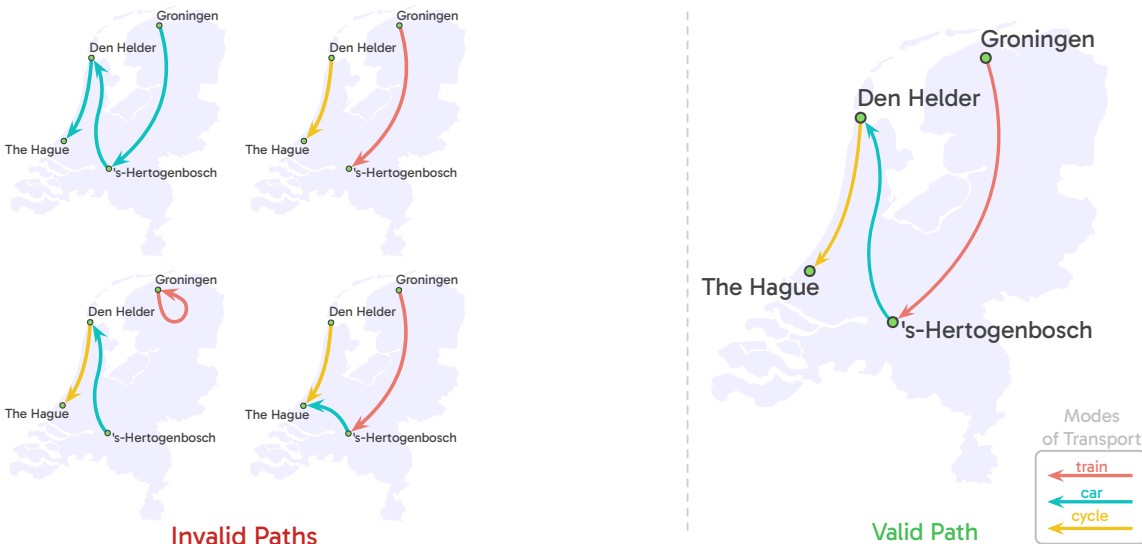

Figure 1: An illustration of Example 1. The task is to plan possible paths that connect four Dutch cities, using a continuous path without revisiting any city, and incorporating each mode of transport: train, car, and cycle. On the left, we show invalid paths which fail to meet the task's criteria, and on the right, we show a valid path which does.

missing links in large KGs by learning scoring functions that assign the correct entities the highest scores (Ruffinelli et al., 2019). These link prediction approaches assume that each link (also known as a *triple*) in a Knowledge Graph can be predicted *independently*. In this view of Knowledge Graphs, each triple is an "atomic fact" that is true or false independent of other triples. We argue that this assumption of the link prediction task prevents it from addressing more complex problems. We illustrate this perspective using Example 1 and 2 as running examples.

**Example 1** (Path Planning). *Depicted in Figure 1 is the task of planning paths that connect four cities in the Netherlands. While there are many potential solutions for this problem, this is a challenging task as it requires understanding the global and local structure of path graphs, as well as learning the underlying constraints governing these graphs, such as having continuous paths, avoiding traversing through the same city more than once, utilising different transportation modes (car, cycle and train), and terminating the path after visiting a certain number of cities.[1] [2] These constraints must be effectively captured for planning valid paths. To predict a segment of the path, a machine learning model would need to consider the previous segments and keep track of the modes of transport already used.*

---

1. Requiring the use of all modes of transport ensures challenging problem-solving process as the solution is not just about finding a path but also about integrating constraints similar to those that would occur in real-world scenarios.
2. We do not impose any constraints on the total travel distance, travel between particular cities, or the need to list out all valid paths.

**Example 2** (Generating Movie Description). *Another task we can consider is generating movie descriptions for KGs. Here, each star-shaped subgraph would describe a movie, with the central node representing a movie and its attributes as peripheral nodes. Each of these attributes can be connected to the central movie node with labelled edges that describe the relationship, such as "has_director", "has_genre", and "has_actor". We need to capture the underlying constraint that every movie needs at least one director, one actor and a movie genre. To generate valid movie descriptions, a machine learning model would need to take into account the structure of the subgraph to make associations between actors, directors and genres, resulting in a star pattern.[3] For instance, a movie directed by Tim Burton is likely to cast Johnny Depp and Helena Bonham Carter and typically gravitates towards the genres of fantasy, gothic, horror, and comedy.*

Link Predictors would solve the problems from Examples 1 and 2 by predicting links of the subgraphs independently of each other (Wen et al., 2016), leading to solutions that are unlikely to satisfy the expressed constraints. In contrast, this paper introduces the task of generating several connected links (or triples) together, modelling their interdependencies. We call this task *subgraph inference*: Instead of predicting missing links, subgraph inference focuses on predicting a set of interdependent links (*subgraphs*).

The lack of datasets with well-understood semantics has made it difficult to determine how well KGE models capture semantics. Existing link prediction datasets commonly used for benchmarking KGE models, such as FB15k-237 (Toutanova and Chen, 2015) and WN18RR (Dettmers et al., 2018), lack well-defined logical constraints to investigate semantics thoroughly. Logical constraints help to verify new data and maintain the logical consistency of knowledge bases, thereby preventing degradation in the quality of subsequent inferential tasks after new triples are added.

Thus, the four main contributions of our work are as follows:

1. **Subgraph Inference.** We define a new task, where the goal is to generate, from a set of training examples, novel subgraphs that follow certain logical rules. We specified new metrics that help empirically evaluate the novelty and logical consistency of newly generated subgraphs.

2. **IntelliGraphs.** We propose five new datasets and describe the underlying semantics using First-Order Logic.

    (a) **Synthetic Datasets.** We generated three synthetic datasets, each designed to capture different levels of semantics.

    (b) **Real-world Datasets.** We extract subgraphs from Wikidata [4] according to simple basic patterns to generate two *real-world* datasets.

3. **KGE baselines.** We designed generative models based on three popular KGE models and show that they cannot effectively capture semantics.

---

3. Our primary focus is on ensuring that the descriptions of these movies adhere to logical constraints, regardless of whether the movies actually exist.

4. https://www.wikidata.org

4. **Data Generator.** We developed a Python package that randomly generates and verifies subgraphs using predefined logical constraints.

The datasets and generators are publicly available on: `https://github.com/thiviyanT/IntelliGraphs`. The generator is available as a Python package which can be installed through PyPI, and Conda package managers.[5] To ensure long-term preservation and easy access, we made the datasets available on Zenodo. [6]

## 2 Benchmark Tasks

In this section, we discuss related benchmark tasks and their limitations, and then, we introduce a novel research task we call *subgraph inference.*

### 2.1 Limitations of Link Predictors

**Binary relations**    KGE models exploit structural regularities to perform Knowledge Graph completion, often neglecting the semantics of the datasets. The last decade has seen the developments of several KGE models (Ruffinelli et al., 2019), which predict the likelihood that a pair of entities are related by a given binary relation. However, a set of binary relations cannot represent an N-*ary* relation because the links depend on each other. Regardless of the context, KGE models assign a set of probabilities on links, and those probabilities are independent of each other.

**N-*ary* relations**    Link prediction has been extended to cover N-*ary* relations, where the goal is to predict a missing link in an N-*ary* fact. N-*ary* relation can operate on any arbitrary number of entities. Modelling N-*ary* relations as triples and treating them as entities in binary relations results in a loss of structural information (Wen et al., 2016). Wen et al. (2016) define N-*ary* relations as the mappings from the attribute sequences to the attribute values, such that each N-*ary* fact is an instance of the corresponding N-*ary* relation. GRAN is a graph-based approach which uses a Transformer decoder to score N-*ary* facts (Wang et al., 2021). NeuInfer uses fully-connected neural networks to embed N-*ary* relations and score candidate triples (Guan et al., 2020). These models were evaluated by inferring an element in an N-*ary* fact.

Because a single N-ary relation can be represented in a set of binary relations (i.e. triples), subgraphs can be used to represent N-ary relations. This means that subgraph models could be used to solve N-ary relation prediction, but the task is strictly broader than that: every single N-ary relation can be represented as a subgraph, but not every class of subgraphs can be naturally captured by a single n-ary relation.

**Link prediction evaluation**    The standard link prediction evaluation framework (Ruffinelli et al., 2019) uses ranking-based evaluation metrics, such as Hits@k and Mean Reciprocal Rank, which do not explicitly check for the semantics of the predicted links. Instead, the evaluation protocol assumes that the underlying semantics can be indirectly validated if a missing link has been correctly predicted. In our work, we set out to *explicitly* check the semantics of newly generated subgraphs.

---

5. PyPI:      `https://pypi.org/project/intelligraphs`      &      Conda:`https://anaconda.org/thiv/intelligraphs`

6. `https://doi.org/10.5281/zenodo.7824818`

## 2.2 Subgraph Inference

A *Knowledge Graph*, $G$, is a tuple $G = (V, E, \mathcal{E}, \mathcal{R}, L)$. $E$ is a set of edges where $E = V \times \mathcal{R} \times V$ and $\mathcal{R}$ is the set of relations. $V$ is set of nodes drawn from the set of possible entities $\mathcal{E}$ in $G$. $L$ is a set of functions that define the semantics of $G$ by determining which structures are permissible or not in $G$.

Given a Knowledge Graph $G$, we call a *subgraph* $F$ a tuple $\left(V^f, E^f, \mathcal{R}\right)$ where $V^f \subset V$ and $E^f = \left\{(u, r, v) \mid u \in V^f, v \in V^f\right\}$, $r \in \mathcal{R}$ and $(u, r, v) \in E$. We require subgraphs to be connected graphs. Every subgraph complies with the semantics of the Knowledge Graph, $L_G$.

For Example 1, a path can be expressed as a subgraph $F$ with the multiple triples, such as `train_to(Groningen, 's-Hertogenbosch)`, `drive_to('s-Hertogenbosch, Den Helder)`, `cycle_to(Den Helder, The Hague)`.[7] The meaning of the triples (or segments of the path) would only make sense collectively, rather than individual set of triples. For the pathfinding the problem, the semantics of the graphs is defined by a set of requirements, such as having a continuous path and no loops, and it can be evaluated by a set of functions that return boolean outputs.

**Problem Statement** *Subgraph Inference* is the task of inferring missing subgraphs given a set of existing subgraphs from a Knowledge Graph, $G$.[8] The inferred subgraphs must adhere to the semantics of the original KG. We define the task as follows: Given a set of known subgraphs $S_G^k$ from a given Knowledge Graph $G$, infer missing subgraphs $S_G^m$ that comply with the logical constraints of the Knowledge Graph, $L_G$, and we assume we have access to $L_G$. At test time, $L_G$ is used to evaluate the semantics of the model output during evaluation.

These subgraphs can be added back to the KG; therefore, this task can be seen as an extension of link prediction. To make this extension complete, we should also specify how the training subgraphs are extracted from $G$. However, to isolate the question of generative modelling of knowledge graphs, we take this process as given in our tasks. For instance, in the two real-world datasets, we extract subgraphs from Wikidata according to a hand-designed pattern. In the synthetic datasets, we simply provide a set of small knowledge graphs over a shared set of entities and relations, leaving the larger graph $G$ entirely implicit. With this choice, the task reduces to training a generative model over small knowledge graphs with a shared set of entities and relations.

**Key Challenges.** IntelliGraphs datasets contain complex semantics, including constraints related to entity types and graph structure. The datasets contain small subgraphs with sparse connections. A machine learning model is required to learn these constraints from a limited set of examples. Furthermore, the semantics of the generated subgraphs need to be verified efficiently at test time.

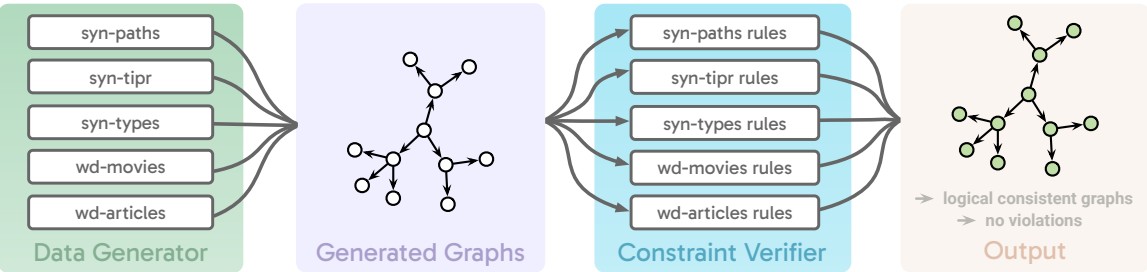

Figure 2: A schematic illustration of the data generation procedure. All generated subgraphs are validated using the constraint verifier to make sure that they do not violate any logical constraints. The data generator and constraint verifier are part of the IntelliGraphs Python package.

## 3 IntelliGraphs

We introduce five new benchmark datasets where each dataset tests different semantics. Figure 2 provides an overview of the data generation and verification process. Table 1 shows key statistics about the synthetic and real-world graphs.

**Data Generator.** For a given dataset, the data generator $D$ uses an incremental construction approach rather than enumerating all valid subgraphs. Subgraphs are built triple-by-triple, with constraints validated at each step. For example, in `syn-paths`, generation begins with a randomly selected source city, then path segments are added sequentially while checking for path continuity and cycle avoidance. This incremental approach is highly efficient, generating approximately 1,000 synthetic subgraphs per second with minimal constraint verification overhead. The random sampling strategy prevents memorizable patterns through three mechanisms: (1) uniform sampling at each construction step, (2) random entity pairing across instances (e.g., "Amsterdam" connects to different cities and transport modes across different samples), and (3) leveraging large configuration spaces (e.g., 49 cities × 3 transport modes yields millions of valid paths). This design forces models to learn logical rules rather than memorizing entity co-occurrences, as the underlying logical constraints are the only learnable pattern across instances. For Wikidata datasets, generation involves extracting subgraphs [9], which takes longer due to graph traversal and entity frequency filtering. However, this is a one-time preprocessing cost, and the resulting datasets are distributed via Zenodo, eliminating regeneration needs for users. In Section 7.4.1 in the Appendix, we describe the algorithm used to generate the five datasets in detail. We ensure that all generated subgraphs $F$ do not violate any logical constraints $L_G$.

**Existential Nodes** In some settings, it is necessary to have nodes that refer to entities that only occur in one instance. For example, in the `wd-movies` dataset introduced below, each subgraph in the data represents a movie. Its actors, directors and genres are entities that

---

8. In symbolic AI, the term *inference* refers to *formal reasoning*. Here we use it to mean estimating the probability distribution over the model's unobserved variables given observed data (*i.e.* probabilistic inference).

9. Subgraphs extracted from a pre-downloaded HDT dump (March 2021).

Table 1: The size of the training, validation and test split for the five datasets used in this work. The number of edges is fixed for the synthetic datasets and is variable for the Wikidata-based graphs.

| Dataset | Split (train/val/test) | Entities | Relations | Edges |
|---------|------------------------|----------|-----------|-------|
| syn-paths | 60000/20000/20000 | 49 | 3 | 3 |
| syn-types | 60000/20000/20000 | 30 | 3 | 10 |
| syn-tipr | 50000/10000/10000 | 130 | 5 | 5 |
| wd-movies | 38267/15698/15796 | 24093 | 3 | $2-21$ |
| wd-articles | 54163/22922/22915 | 60932 | 6 | $4-212$ |

occur in multiple instances, so a model can learn representations by observing the different contexts in which they occur. However, each instance also contains one node representing the movie the graph describes. These only occur in one instance, so a model cannot learn a representation for the specific instance, only a general representation which expresses that some movie exists for which this subgraph is true. We call such nodes *existential nodes* (by analogy to existentially quantified variables in logical formulas) and use a special label, such as `_movie`, to refer to them in all instances.[10] Strictly speaking, this turns the predicted subgraphs into subgraph *patterns* of the Knowledge Graph $G$, but we refer to them as subgraphs to keep the terminology simple.

### 3.1 Semantics

We use First-Order Logic (FOL) to express the underlying logical constraints of the datasets. These logical constraints, $L$, were hand-crafted for every dataset, and we ensured that all subgraphs in the training, validation and test split complied with the logical rules. Section 7.6 (in the Appendix) provides a complete set of logical constraints for each IntelliGraphs dataset.

**Logical Constraint Verifier.** The *Constraint Verifier* is a set of functions $V$ that verifies whether the set of logical constraints $L$ hold in a generated subgraph $F$. We provide a logic constraint verifier for each dataset within the IntelliGraphs Python package.[11] The logical constraint verifier $V(F, L)$ returns true if and only if the subgraph $F$ is consistent with all logical rules $L$.

---

10. For most models, the difference will only be in the interpretation. For example, our baseline models will learn one embedding vector for the node labelled `_movie`, which we use wherever movies occur. As such, we do not treat it differently from the node labelled `Antonio_Banderas`, although when we interpret the graph, these nodes mean different things.

11. A reasoning engine could also be used for checking the subgraphs for logical consistency. We wrote a set of functions in Python for constraint verification and embedded it into the IntelliGraphs Python package to easily verify graphs without loading them into a reasoning engine.

### 3.2 Synthetic Datasets

Synthetic datasets allow complete control over the problem setup and provide a convenient testbed for developing new machine learning models. The dataset is generated by the generator, $D$. We checked if the generated subgraphs satisfy the logical rules $L$.[12] Here is a brief description of the synthetic datasets:

- `syn-paths` is a dataset with path graphs with Dutch cities representing the nodes and edges representing the route and mode of transport. This dataset was inspired by Example 1. The requirements are the following: 1) it must be continuous paths, 2) a path cannot travel through the same city more than once, 3) different transportation modes must be utilised for each segment of the path (car, cycle and train), and 4) the path must come to terminate after visiting four cities.

- `syn-types` contains entities with types: languages, countries, and capital cities. These are connected by three relations according to the relation's type constraints: `same_type_as` can only exist between the same entity types, `could_be_part_of` between a capital city and a country, and `could_be_spoken_in` between a language and a country.[13]

- `syn-tipr` contains subgraphs based on the *Time-Indexed Person Role* (tipr) graph pattern describing a fictional person.[14] Here, the semantics are defined by the *tipr* graph pattern, ensuring that an individual has both a name and role as well as a time index. Additionally, the beginning of a time interval must precede its ending.

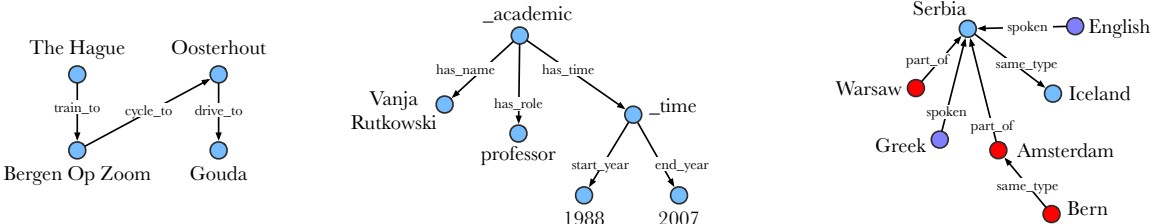

Figure 3: Example subgraphs from the three synthetic datasets. **Left:** `syn-paths` shows a valid path through four Dutch cities using three different transport modes (train, cycle, drive). **Center:** `syn-tipr` shows a time-indexed person role describing an academic's position with temporal constraints (start year must precede end year). **Right:** `syn-types` shows type-constrained relations between entities of three types: countries, cities and languages.

---

12. It is important to note that logical consistency does not equate to factual accuracy. We simply want to ensure that the generated dataset is consistent with the logical rules.
13. The connections between the entities are chosen arbitrarily, ignoring the real-world relations between the entities. However, they do conform to predefined logical constraints.
14. `http://ontologydesignpatterns.org/wiki/Submissions:Time_indexed_person_role`

### 3.3 Real-World Datasets

Wikidata (Vrandečić and Krötzsch, 2014) is a large graph-structured knowledge base which consists of crowdsourced factual knowledge on various topics. We created two datasets from Wikidata using specific graph patterns to extract subgraphs about movies and research articles. Unlike the synthetic datasets, the size of the subgraphs in these datasets can be variable. Here is a brief description of the two datasets:

- `wd-movies` contains small graphs extracted from Wikidata that describe movies. This dataset was inspired by Example 2. Each subgraph contains one existential node representing the movie, entity nodes for the movie's director(s) connected by a `has_director` relation, entity nodes for the movie's cast connected by a `has_actor` relation and an entity for the movie's genre connected by a `has_genre` relation.

- `wd-articles` contains small graphs that describe research articles extracted from Wikidata. Each article is annotated by an ordered list of authors, implemented by a blank node for each author linked to a node representing the author and to a node representing the order in the author list. We add a list of the other articles that the current article references, and a list of subjects, together with selected superclasses of those subjects. In this dataset, most node types, including the article's node, may be existential or entity nodes.

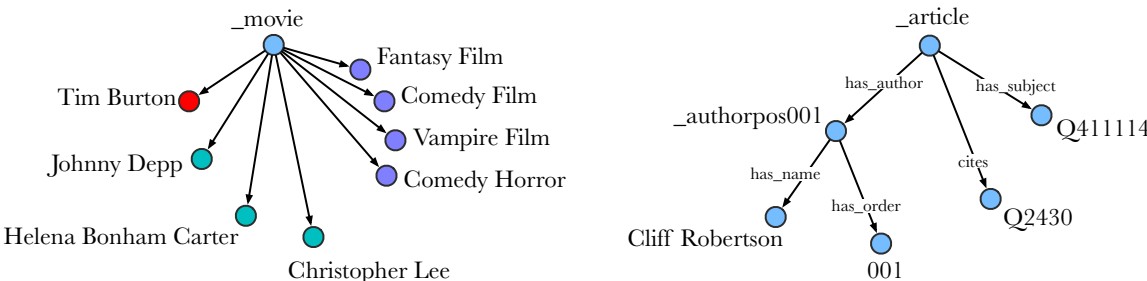

Figure 4: Example subgraphs from the two Wikidata-based datasets. **Left:** `wd-articles` shows a research article with an ordered author (linked via an intermediate node storing name and position), citations to other articles, and subject annotations. **Right:** `wd-movies` shows a star-shaped movie description with director, cast members, and genres. Dashed nodes (e.g., `_movie`, `_article`, `_authorpos001`) represent existential entities that are unique to each subgraph instance.

## 4 Evaluation

### 4.1 Baseline Models

To the best of our knowledge, no probabilistic models in the literature can infer new subgraphs for knowledge graphs. Therefore, we developed a set of simple baselines inspired by traditional

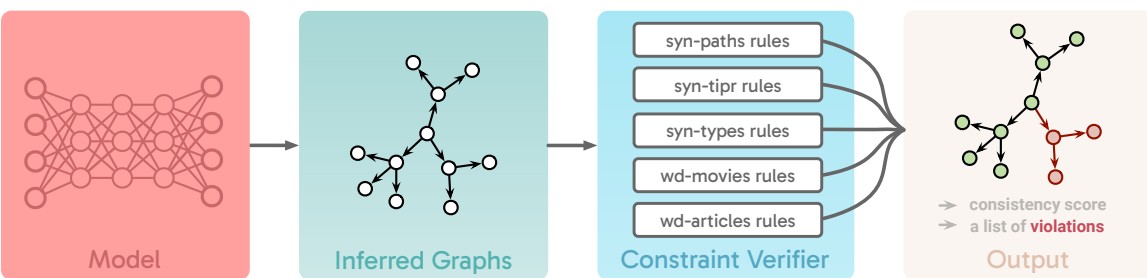

Figure 5: A schematic illustration of the evaluation procedure for the IntelliGraphs dataset. The Constraint Verifier checks the graphs generated by a machine learning model to check for any potential logical violations and computes a *consistency score.*

KGE models: ComplEx (Trouillon et al., 2016), DistMult (Yang et al., 2014) and TransE (Bordes et al., 2013). Traditional KGE models are trained to rank all possible triples to give the correct triple the highest score (Ruffinelli et al., 2019). ComplEx, DistMult and TransE all use different scoring functions. TransE represents relations as translation between entities, whereas DistMult models relations as bilinear interactions. ComplEx extends DistMult using complex-valued embeddings.

We focused on KGE models as they represent the dominant paradigm in Knowledge Graph. The three models we selected (TransE, DistMult, and ComplEx) span different representational paradigms: translational embeddings, bilinear interactions, and complex-valued embeddings respectively. One might consider graph generative models such as GraphRNN (You et al., 2018) or GraphVAE (Simonovsky and Komodakis, 2018) as alternative baselines. However, these architectures were designed for homogeneous graphs rather than KGs with typed edges and entity-relation structure. Adapting them would require non-trivial modifications: redefining generation to handle $(subject, predicate, object)$ triples, incorporating entity and relation type constraints, and modifying their generation mechanisms to respect logical constraints. Including models without careful adaptation for KGs could conflate two questions: *are poor results due to the fundamental challenge of constraint-aware generation, or due to suboptimal adaptation choices?* By focusing on well-understood KGE baselines with principled adaptations, we ensure the evaluation clearly reveals the core challenges. We believe IntelliGraphs will inspire the development of novel architectures specifically designed for constraint-aware knowledge graph generation, rather than forcing adaptations of models built for fundamentally different tasks.

We model a subgraph $F$ by decomposing it into its entities and structure $F = (E, S)$, that is, $p(F) = p(S \mid E) \, p(E)$. Unlike traditional KGE models, we train our baseline models with a maximum likelihood objective.

We decompose the objective function as follows:

$$-\log_2 p(F) = -\log_2 p(S|E) - \log_2 p(E). \tag{1}$$

Each of the terms in Equation 1 can be read as separate codelengths: $-\log_2 P(E)$ describes the bits required to encode the entities, and $-\log_2 P(S \mid E)$ describes the bits required to encode the structure once the entities are known.

We model $p(E) = \prod_{e \in E} p(e)$, with $p(e)$ estimated as the relative frequency of $e$ in the training data (the proportion of training subgraphs it occurs in). We train KGE models to estimate $p(S \mid E)$. We use

$$p(S \mid E) = \prod_{(s,p,o) \in S_T} p((s,p,o) \mid E) \prod_{(s,p,o) \in S_N} 1 - p((s,p,o) \mid E), \tag{2}$$

where $S_T$ represents the triples in the subgraph $F$, and $S_N$ represents all possible triples that are not in the subgraph (*i.e.* all possible *negatives*).

**Uniform Baseline.** Our *random* baseline model generates a random graph prediction by sampling $p(E)$ and $p(S|E)$ from a uniform distribution. The uniform baseline assumes that all possible entities in $E$ and all possible edges in $S$ given $E$ are equally likely to be sampled. We also encode certain properties of the datasets, such as the fact that self-loops are not allowed (see Section 7.1.2 in the Appendix). It then computes the exact number of bits required to represent a graph for the datasets using $-log_2(p)$. This model does not need to be trained. This baseline serves as a naive model to estimate the upper bounds on the compression or information content in a dataset, providing a benchmark against which more sophisticated models can be compared.

## 4.2 Evaluation by bits-per-graph

The most common objective for a generative model is probably maximum likelihood: the probability of a graph from the test data under the model should have maximal probability, or, equivalently, minimal negative log probability. When base 2 logarithms are used, the latter quantity, $-\log_2 p(S, E)$, can be interpreted as the number of bits required to compress the graph (Rissanen, 1978; Grünwald, 2007). Averaging over all graphs, we arrive at a metric of *bits-per-graph* to evaluate how well our model satisfies the maximum likelihood objective.

## 4.3 Semantics

Figure 5 provides an overview of the evaluation procedure. We evaluate the semantics of graphs generated by our baseline models using the following evaluation metrics: 1) *% Valid Graphs* is the probability of sampling graphs that are logically valid according to the logical constraints for each dataset, 2) *% Novel Graphs* is the probability of sampling graphs that are not in the training data, 3) *% Novel & Valid Graphs* is the probability of sampling graphs that are logically valid and are not in the training data, and 4) *% Empty Graphs* represents the likelihood of generating samples that result in no graphs, which can occur if either $p(E)$ or $p(S \mid E)$ is too low. An ideal model gives a high probability of sampling logically valid graphs but uses a minimal number of code lengths to compress graphs.

**Traditional Link Prediction.** Traditional link prediction metrics such as Mean Reciprocal Rank (MRR) and Hits@K evaluate the ranking of individual triples against corrupted alternatives, assuming independence between predictions. These metrics are fundamentally incompatible with subgraph inference for several reasons. First, the space of possible subgraphs is combinatorially large (exponential in the number of triples), making

ranking-based evaluation intractable. Second, subgraphs are either valid or invalid with respect to constraints, a binary outcome rather than a rankable quantity. Third, partial correctness cannot be captured by ranking individual triples when semantic validity emerges from their interconnections. Consider transitivity as an example: if a valid subgraph requires triples $(A \rightarrow B)$, $(B \rightarrow C)$, and $(A \rightarrow C)$, a model might correctly predict the first two with high confidence (scoring well on MRR) while failing to infer the third, producing an invalid subgraph. The semantic validity of the whole cannot be decomposed into independent predictions. Our proposed metrics, validity rate and constraint satisfaction breakdown, directly measure whether models can generate semantically valid knowledge structures.

## 4.4 N-ary Link Prediction

The simplest tasks in IntelliGraphs can be modelled directly as an N-ary link prediction problem. For instance, the `syn-paths` graph is a 4-ary hypergraph, and predicting hyperlinks on this graph is one way to solve the problem. However, as the tasks increase in complexity, the limitations of N-ary link prediction become clear. To model a task like `wd-articles` purely with link prediction, a single n-ary relation would need to capture the entirety of one subgraph describing an article, its author in order, the subjects of the article, and other features of the subgraph. Moreover, for the more complex tasks, the subgraph size is variable, which means that a single N-ary relation cannot capture the entire subgraph unless the arity of the relation is somehow made variable. We leave the empirical study for future work.

Table 2 shows that the KGE baselines generally result in higher code lengths than the random baseline. Among the KGE models, ComplEx tends to achieve lower total code lengths, suggesting it is more effective at modeling the structure of these graphs $p(S|E)$, despite its use of complex-valued embeddings. The scale of complexity, represented by code length, increases rapidly from synthetic to real-world datasets. For instance, the highest code length for the synthetic dataset `syn-tipr` is 69.51 bits (for the TransE baseline), while the lowest code length for the real-world dataset `wd-movies` is 202.68 bits (for the ComplEx baseline). The datasets `wd-movies` and `wd-articles` have many more entities to sample, making them more challenging to compress.

**Interpreting the Random Baseline.** The random baseline achieving lower code lengths than trained KGE models may seem counterintuitive, but this result is actually revealing. The random baseline computes the *theoretical minimum* bits required under a uniform distribution assumption (see Section 7.1.2). When KGE models, despite training on the data, fail to achieve lower code lengths than this uniform baseline, it indicates they are not learning more efficient representations of the semantic structure. In information-theoretic terms, KGE models are not capturing the logical regularities that would enable compression. A model that truly understands the constraints should achieve much lower code length by exploiting these patterns. The seemingly competitive random baseline performance actually underscores how poorly current models capture semantic structure—they learn no more about the data distribution than a model that assumes everything is equally likely.

## 4.5 Subgraph Inference

Table 3 shows the probabilities of sampling graphs that are logically consistent. We perform subgraph inference under two different settings:

Table 2: Estimate of the codelengths, $-\log_2 p(F)$, (the number of bits) required to compress a graph using the four baseline models for IntelliGraphs datasets. We used the test split for this.

| Datasets | Baseline Models | $-\log_2 p(S\|E)$ | $-\log_2 p(E)$ | $-\log_2 p(F)$ |
|---|---|---|---|---|
| | *random* | 12.80 | 17.69 | 30.49 |
| | TransE | 16.19 | 33.69 | 49.89 |
| syn-paths | DistMult | 14.90 | 33.69 | 48.58 |
| | ComplEx | 20.71 | 33.69 | 54.39 |
| | *random* | 29.14 | 32.47 | 61.61 |
| | TransE | 28.70 | 40.81 | 69.51 |
| syn-tipr | DistMult | 26.70 | 40.81 | 67.51 |
| | ComplEx | 23.15 | 40.81 | 63.96 |
| | *random* | 16.84 | 19.18 | 36.02 |
| | TransE | 19.05 | 29.21 | 48.26 |
| syn-types | DistMult | 18.24 | 29.21 | 47.46 |
| | ComplEx | 18.48 | 29.21 | 47.69 |
| | *random* | 53.86 | 117.74 | 171.60 |
| | TransE | 51.39 | 157.21 | 208.60 |
| wd-movies | DistMult | 51.29 | 157.21 | 208.50 |
| | ComplEx | 45.46 | 157.21 | 202.68 |
| | *random* | 295.60 | 398.20 | 693.80 |
| | TransE | 280.67 | 629.98 | 910.65 |
| wd-articles | DistMult | 271.94 | 629.98 | 901.91 |
| | ComplEx | 257.33 | 629.98 | 887.30 |

- **Sampling $P(E)$ and $P(S \mid E)$.** In this setting, the baseline models independently sample both the set of entities $E$ relevant for a subgraph and then predict the edge connectivity $S$ among these entities. Our results indicate that the probability of sampling valid graphs using this method is consistently 0%. This is because when entities are sampled independently, it is highly unlikely that the selected set can be connected into a valid graph that satisfies the dataset's logical constraints. Selecting incorrect entities negatively impacts the structure prediction. This highlights that this task is challenging.

- **Sampling only $P(S \mid E)$.** In this setup, the model is given the correct set of entities (*i.e.*, nodes $E$ randomly chosen from test graphs) and only needs to predict the edge connections among them. Here, the baseline model effectively becomes a link predictor. Despite this advantage, the baseline models generated few logically consistent subgraphs, reflecting the datasets' complexity and the limitations of KGE models in learning semantics. While most KGE models generated some valid path graphs, syn-tipr, requiring temporal reasoning, remained challenging for all models. Inferring correct entity types in syn-types was possible for a small number of subgraphs.

Table 3: Semantic validity of the graphs produced by our baseline models. High values for *% Novel & Valid Graphs* are desirable. We have tested subgraph inference under two settings: 1) Sampling from *both* $P(E)$ and $P(S \mid E)$, and 2) Sampling from $P(S \mid E)$ *only*, taking $E$ from the test data. We check the novelty of the sampled graphs by comparing them against the training and validation set. We used the same hyperparameters from the model compression experiments here. The best performing models for each dataset is **bolded**.

| Setting | Dataset | Model | % Valid Graphs | % Novel & Valid Graphs | % Novel Graphs | % Empty Graphs |
|---|---|---|---|---|---|---|
| Sampling from $P(E)$ and $P(S \mid E)$ | syn-paths | random | 0 | 0 | 100 | 0 |
| | | TransE | 0.25 | 0.25 | 23.45 | 76.55 |
| | | DistMult | 0.69 | 0.69 | 14.59 | 85.41 |
| | | ComplEx | 0.71 | 0.71 | 14.27 | 85.73 |
| | syn-tipr | random | 0 | 0 | 100 | 0 |
| | | TransE | 0 | 0 | 5.58 | 94.42 |
| | | DistMult | 0 | 0 | 13.34 | 86.66 |
| | | ComplEx | 0 | 0 | 4.95 | 96.05 |
| | syn-types | random | 0 | 0 | 100 | 0 |
| | | TransE | 0.21 | 0.21 | 15.44 | 84.56 |
| | | DistMult | 0.13 | 0.13 | 12.46 | 87.53 |
| | | ComplEx | 0.07 | 0.07 | 10.25 | 89.75 |
| | wd-movies | random | 0 | 0 | 100 | 0 |
| | | TransE | 0 | 0 | 14.61 | 85.39 |
| | | DistMult | 0 | 0 | 12.93 | 87.07 |
| | | ComplEx | 0 | 0 | 1.87 | 98.13 |
| | wd-articles | random | 0 | 0 | 100 | 0 |
| | | TransE | 0 | 0 | 4.58 | 95.42 |
| | | DistMult | 0 | 0 | 0 | 100.00 |
| | | ComplEx | 0 | 0 | 2.46 | 97.54 |
| Sampling from $P(S \mid E)$ only | syn-paths | random | 0 | 0 | 100 | 0 |
| | | TransE | 5.25 | 5.25 | 95.52 | 4.48 |
| | | DistMult | 9.69 | 9.69 | 95.28 | 4.71 |
| | | **ComplEx** | **10.10** | **10.10** | **95.58** | **4.42** |
| | syn-tipr | random | 0 | 0 | 100 | 0 |
| | | TransE | 0 | 0 | 99.45 | 0.55 |
| | | DistMult | 0 | 0 | 99.43 | 0.57 |
| | | ComplEx | 0 | 0 | 99.64 | 0.36 |
| | syn-types | random | 0 | 0 | 100 | 0 |
| | | TransE | 1.43 | 1.43 | 95.42 | 4.58 |
| | | **DistMult** | **1.44** | **1.44** | **96.19** | **4.81** |
| | | ComplEx | 1.01 | 1.01 | 94.17 | 5.83 |
| | wd-movies | random | 0 | 0 | 100 | 0 |
| | | TransE | 0.07 | 0.07 | 97.01 | 2.99 |
| | | DistMult | 0.10 | 0.10 | 95.86 | 4.17 |
| | | **ComplEx** | **0.41** | **0.41** | **93.04** | **6.96** |
| | wd-articles | random | 0 | 0 | 100 | 0 |
| | | TransE | 0 | 0 | 98.35 | 1.65 |
| | | DistMult | 0 | 0 | 98.77 | 1.23 |
| | | ComplEx | 0 | 0 | 100.00 | 0.00 |

## 5 Related Work

**Datasets for Query Embedding.** Query Embedding (QE) involves interpreting complex logical queries, commonly represented as a small graph, and evaluated on QE datasets, such as GQE (Hamilton et al., 2018), Query2Box (Ren et al., 2020), and BetaE (Ren and Leskovec, 2020). Ren et al. (2023) presents a comprehensive comparison of datasets. As Ren et al. (2023) highlight in their recent survey, query embedding datasets lack logical rules and types. Although the datasets in IntelliGraphs are similar to query embedding datasets, there is a difference in the purpose and applications. Our datasets could be used for learning distributions to infer new logically consistent subgraphs. In contrast, QA datasets are concerned with reasoning using logical rules to find a missing entity.

**Datasets for n-*ary* Relations.** N-ary relations are relations involving more than two entities. Various methods have been studied in the literature that embed complex N-ary relations, often in non-euclidean spaces (Wang et al., 2021; Wen et al., 2016). The difference between N-*ary* relations and subgraphs is explained in Section 2.1.

**Datasets for Neurosymbolic learning and reasoning.** If we interpret knowledge graphs as a set of logical statements, we can see that the task of subgraph prediction is a neurosymbolic method: it combines symbolic systems with neural networks. Datasets have been proposed to test various aspects of such systems: interpretability, reasoning, and generalization capabilities. Several datasets were proposed to evaluate the understanding and reasoning of complex rules and abstract concepts. Table 4 (in the appendix) compares different datasets for Neurosymbolic AI from the literature. Existing datasets focus primarily on the image and text modalities, neglecting background knowledge expressed in graphs.

## 6 Conclusion

Existing KG datasets used for representation learning lack well-understood semantics, which limits studying how well KGE models capture new semantics. In our work, we propose *Subgraph Inference* as a new research problem and present *IntelliGraphs*, a collection of five new datasets for benchmarking models. Furthermore, we used baseline models inspired by traditional KGE models to estimate the code lengths of these graphs and sample logically valid subgraphs. Our findings show that traditional KGE models show a limited understanding of semantics after training. We observed a rapid increase in complexity, represented by code lengths, from synthetic to real-world datasets. This complexity makes real-world datasets more challenging to compress, which is an essential consideration for future research in graph compression. We found that the probability of sampling valid graphs was consistently low, emphasizing the complexity and difficulty of the task.

**Limitations.** *Subgraph inference* assumes that the semantics of a KG is known. However, in some cases, this assumption may not hold. Furthermore, our datasets assume we test the machine learning models in a transductive setting; entities and relations not seen during training will not be handled well.

**Scalability Considerations.** The primary focus of IntelliGraphs is on establishing a benchmark that reveals the semantic limitations of current methods, not on optimizing for scale. Our dataset sizes were deliberately chosen to allow thorough experimental validation and ensure reproducibility on standard academic hardware. It is important to note that

constraint checking only occurs during evaluation, not during training, making it a one-time computational cost that can be easily parallelized across generated subgraphs. The constraint verification scales linearly with the number of subgraphs and can be distributed across machines, if needed.

**Applications of IntelliGraphs.** It is imperative to have guarantees for safety-critical applications to prevent machine learning models from making fatal mistakes. To develop these systems, datasets with logical constraints are helpful. In some problem domains, there is little or no data available such as cases where training machine learning models on sensitive data for medical or industrial use cases. While we do not provide datasets that are directly applicable to these use cases, IntelliGraph's dataset generation framework can be used to generate synthetic datasets using background knowledge about the problem domain.

**Ethics Statement.** Our synthetic graphs are based on hand-crafted logical constraints (see the Appendix) and thus, we discourage usage in applications where factual accuracy matters. However, `wd-movies` and `wd-articles` are based on real-world factual knowledge retrieved from Wikidata, and therefore, certain biases may be inherited from Wikidata. Since these datasets are likely unsuitable for training production models, we do not expect that these biases will ever affect systems making real-world decisions.

`wd-movies` contains a small number of sensitive movie genres, such as *rape and revenge film*, *pornographic film* or *erotic film*. We refrained from filtering out movies with these genres, taking the view that Wikidata's norms for curating data from a neutral point of view are sufficiently well-considered that they should carry over to our datasets. Practitioners should nevertheless be aware of these genres and may want to filter them out when generating samples for publication.

Transparency about dataset creation and maintenance is critical for adopting new machine learning datasets (Gebru et al., 2021). We provide a data card for IntelliGraphs to provide further information about the datasets.

# Acknowledgments and Disclosure of Funding

**Acknowledgements**   We would like to thank Frank van Harmelen and Patrick Koopmann for their feedback on this work.

# Broader Impact Statement

The datasets from IntelliGraphs are highly specialized, and were designed to evaluate specific aspects of reasoning. Thus, the datasets do not possess any transferable value beyond academic or industrial research. However, a system that is capable of solving these problems could pose a risk to the broader technological landscape, potentially facilitating unintended applications, such as mass surveillance.

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

# 7 Appendix

## Contents

## 7.1 Baseline Details

### 7.1.1 Probabilistic KGE

The *probabilistic* baselines estimate $p(E)$ and $p(S|E)$ by using trained Knowledge Graph Embedding (KGE) models, based on ComplEx (Trouillon et al., 2016), TransE (Bordes et al., 2013) and DistMult (Yang et al., 2014). This approach uses the learned embeddings and observed frequencies from the training data to assign probabilities to entities and edges, allowing for a more informed estimation of the bits required to encode subgraphs compared to the *random* baseline (described in Section 7.1.2).

**Entity Probabilities** $p(E)$**.** For synthetic datasets, we assume a fixed number of nodes and edges in the subgraphs. We compute the probability of each entity $e$ based on its relative frequency in the training data, applying a Laplace smoothing:

$$p(e) = \frac{\text{freq}(e) + \lambda}{\sum_{e' \in \mathcal{E}} \text{freq}(e') + (\lambda \cdot |\mathcal{E}|)},$$

where freq(e) is the count of the total entities $e$ appears in the subgraphs in the training set, $\lambda$ is the smoothing parameter (we set $\lambda$ to $1 \times 10^{-4}$), and $|\mathcal{E}|$ is the total number of entities. The bits required to encode the entities in a subgraph $E$ are calculated as:

$$\text{Bits}_{p(E)} = \sum_{e \in E} -\log_2 p(e).$$

This calculation assumes that the entities are sampled independently according to their (smoothed) frequencies in the training data.

**Structure Probabilities $p(S|E)$.** Given the entities $E$ in the subgraph, we use a KGE model to estimate the probability of each possible edge between entities in the subgraph in the following steps:

1. **Generate Candidate Triples:**

   We consider all possible triple combinations $(s, r, o)$, where $s$ and $o$ are entities in a subgraph $E_S$, and $r$ spans all possible relations $\mathcal{R}$:

   $$\mathcal{T} = \{(s, r, o) \mid s, o \in E_S, \ r \in \mathcal{R}\}.$$

   This accounts for all possible edges between nodes in a subgraph, considering different types of relations.

2. **Compute Edge Probabilities:**

   The KGE model provides a score $f(s, r, o)$ for each triple in $\mathcal{T}$, which is converted to probabilities using the sigmoid function:

   $$p(s, r, o) = \sigma(f(s, r, o)) = \frac{1}{1 + e^{-f(s,r,o)}}.$$

3. **Calculate Bits for Observed and Non-Observed Edges:**

   For each edge $(s, r, o)$ present in the subgraph $S$:

   $$\text{Bits}_{\text{obs}} = \sum_{(s,r,o) \in S} -\log_2 p(s, r, o).$$

   For edges not present in $S$ (i.e., $(s, r, o) \in \mathcal{T} \setminus S$):

   $$\text{Bits}_{\text{non-obs}} = \sum_{(s,r,o) \in \mathcal{T} \setminus S} -\log_2 [1 - p(s, r, o)].$$

   The total bits required to encode the subgraph structure $S$ given the entities $E$ is:

   $$\text{Bits}_{p(S|E)} = \text{Bits}_{\text{obs}} + \text{Bits}_{\text{non-obs}}.$$

The total compression bits per subgraph is simply the sum of $\text{Bits}_{p(E)}$ and $\text{Bits}_{p(S|E)}$.

**Wikidata Datasets** For Wikidata datasets, the number of edges and nodes varies across subgraphs in the training data. We report the average compression in bits over all graphs. To compute this, we include a term for the maximum possible number of nodes to encode the number of nodes in each graph:[15]

$$\text{Average Bits}_{p(E)} = \frac{1}{N} \sum_{i=1}^{N} \left[ \log_2 n_{\text{max nodes}} + \sum_{e \in E_i} - \log_2 p(e) \right],$$

where $N$ is the total number of graphs, $E_i$ is the set of entities in graph $i$, and $n_{\text{max nodes}}$ is the maximum number of nodes observed across all data splits.

Similarly, for $p(S|E)$, we compute:

$$\text{Average Bits}_{p(S|E)} = \frac{1}{N} \sum_{i=1}^{N} \left[ \text{Bits}_{\text{obs}}^{(i)} + \text{Bits}_{\text{non-obs}}^{(i)} \right],$$

where $\text{Bits}_{\text{obs}}^{(i)}$ and $\text{Bits}_{\text{non-obs}}^{(i)}$ are the bits for observed and non-observed edges in graph $i$, respectively.

**Implementation Details** We apply Laplace smoothing with a small parameter ($\lambda$) to ensure that the entities with zero frequency in the training data are assigned a non-zero probability. This prevents issues with taking the logarithm of zero probabilities. Entities are assumed to be sampled independently with replacement based on their probabilities $p(e)$. While this may overestimate the bits required for synthetic datasets where entities are sampled without replacement, it provides a reasonable approximation for real-world datasets like Wikidata, where the entity distribution may be skewed and the number of possible entities is large. When generating candidate triples $\mathcal{T}$, self-loops (edges where $s = o$) are excluded, giving the baseline models a slight competitive advantage over the random baseline.

### 7.1.2 RANDOM BASELINE

The random baseline model does not require training and provides a baseline by sampling from uniform probability distributions for entities $p(E)$ and graph structure $p(S|E)$.

For synthetic datasets, where we assume a fixed number of nodes and edges in the subgraphs, the number of bits required is calculated as follows:

$$\text{Bits for } p(E) = \log_2 \binom{n_{\text{entities}}}{n_{\text{nodes}}},$$

where $\binom{n_{\text{entities}}}{n_{\text{nodes}}}$ represents the binomial coefficient, i.e., the number of ways to choose $n_{\text{nodes}}$ from $n_{\text{entities}}$ possible entities. This assumes that entities are sampled without replacement. Likewise, we define

$$p(S|E) = \log_2 \binom{n_{\text{possible edges}}}{n_{\text{edges}}}.$$

---

15. This is equivalent to placing a uniform distribution $U(0, n_{\text{max nodes}})$ on the size $n$ of the graph: $-\log p_U(n) = -\log \frac{1}{n_{\text{max nodes}}} = \log n_{\text{max nodes}}$.

Here, $n_{\text{possible edges}} = (n_{\text{nodes}}^2 - n_{\text{nodes}}) \times n_{\text{relations}}$ accounts for all possible edges between nodes, considering the different types of relations, and excluding self-loops. This calculation also assumes that edges are sampled without replacement.

For Wikidata datasets, where the number of edges and nodes varies across subgraphs, we compute the compression bits by averaging the number of bits across all graphs. For simplicity, the maximum number of nodes ($n_{\text{max nodes}}$) and maximum number of edges ($n_{\text{max edges}}$) are taken from the union of the train, validation, and test splits. The bits for $p(E)$ are computed as an average across all graphs and include a term for the maximum possible number of nodes:

$$\text{Average Bits for } p(E) = \frac{1}{N} \sum_{i=1}^{N} \left[ \log_2(n_{\text{max nodes}}) + \log_2 \binom{n_{\text{entities}}}{n_{\text{nodes}}^i} \right],$$

where $n_{\text{nodes}}^i$ is the number of nodes in graph $i$, and $N$ is the total number of graphs. The first term can be read as encoding the number of nodes in the graph (a choice from a set of $n_{\text{max nodes}}$ options) and the second term encodes the entities given this information.

Likewise, for $p(S|E)$:

$$\text{Average Bits for } p(S|E) = \frac{1}{N} \sum_{i=1}^{N} \left[ \log_2(n_{\text{max edges}}) + \log_2 \binom{n_{\text{possible edges}}^i}{n_{\text{edges}}^i} \right],$$

where $n_{\text{possible edges}}^i = n_{\text{nodes}}^i \times (n_{\text{nodes}}^i - 1) \times n_{\text{relations}}$ is computed individually for each graph $i$, assuming edges are sampled without replacement. The term $\log_2(n_{\text{max edges}})$ accounts for the maximum number of edges observed across all splits.

### 7.1.3 SIMPLE GENERATIVE MODEL

To investigate whether architectures capable of modeling triple interdependencies can better capture subgraph-level structure, we developed a simple Variational Autoencoder where the encoder and decoder are Multi-layer Perceptron networks. Unlike KGE models that predict triples independently, a VAE can learn a latent representation that encodes the global structure of subgraphs, jointly modeling the dependencies between edges and nodes.

The VAE follows the same decomposition principle $p(F) = p(S \mid E)\, p(E)$. The model consists of an MLP encoder that maps node and edge embeddings to a latent representation $z$, and an MLP decoder that reconstructs node embeddings and predicts graph structure. Unlike traditional KGE models that focus on triple scoring, MLP-VAE optimizes a variational objective:

$$\mathcal{L}_{VAE} = \mathbb{E}_{q(z|F)}[\log p(S, E|z)] - D_{KL}(q(z|F)||p(z)), \tag{3}$$

where the first term represents reconstruction quality (both entity prediction, $p(E)$, and subgraph structure, $p(S|E)$), and the second term is the Kullback–Leibler divergence between the approximate posterior $q(z|F)$ and the prior $p(z)$.

We performed experiments with the VAE only on the `syn-paths` dataset. While `syn-paths` is structurally simpler than other datasets, its path-connectivity pattern allows clear assessment of a model's ability to capture basic graph semantics. The VAE achieves a compression rate of 37.59 bits per graph for the `syn-paths` dataset, which is better

than the KGE baselines (48.58–54.39 bits) and closer to the random uniform baseline (30.49 bits). This improvement comes from more efficient modeling of both entity distributions and structural relationships, further demonstrating that KGE-based approaches are insufficient to capture the interdependencies between triples.

However, despite the improved compression performance, the VAE also failed to produce any semantically valid graphs when sampling. Developing robust generative models that can reliably produce semantically valid knowledge graph subgraphs remains an open problem and is beyond the scope of this work. The IntelliGraphs benchmark provides a foundation for future research in this direction.

## 7.2 Datasets for Neurosymbolic Methods

Neurosymbolic methods aim to combine neural networks with symbolic representations. Several datasets already exist in the literature for evaluating the performance of neurosymbolic methods. Table 4 highlights widely used datasets used for benchmarking neurosymbolic systems.

## 7.3 Reproducibility Statement

To make our work fully reproducible, we make the codebase of our experiments public and open. Our code is available on `https://github.com/thiviyanT/IntelliGraphs`. For each experiment, we also provide the hyperparameter configurations we used. Furthermore, we have released a new Python package for interacting with the IntelliGraphs datasets through the following software package repositories: **conda** (`https://anaconda.org/thiv/intelligraphs`) and **pypi** (`https://pypi.org/project/intelligraphs`). To ensure long-term preservation and easy access, we made the datasets available on Zenodo (`https://doi.org/10.5281/zenodo.7824818`). Experimental details can be found in the next Section.

## 7.4 Experimental Details

We used the PyTorch library [16] to develop and test the models. All experiments were performed on a single-node machine with an Intel(R) Xeon(R) Gold 5118 (2.30GHz, 12 cores) CPU and 64GB of RAM, with four NVIDIA RTX A4000 GPUs (16GB of VRAM). We used PyTorch's GPU acceleration for training the models. We used the Adam optimiser with variable learning rates (Kingma and Ba, 2014).

### 7.4.1 Hyperparameters

For each dataset, we performed hyperparameter sweeps using every baseline model (TransE, DistMult, ComplEx) using Weights&Biases [17]. For this, we used a random search strategy with the goal of finding the hyperparameter configurations that yield the minimum compression bits on the validation set. We do not include the reciprocal relation model, and we used the highest batch size that we could fit in memory. Table 5 shows the hyperparameter values

---

16. `https://pytorch.org/`
17. `https://wandb.ai/`

Table 4: Brief comparison of commonly used datasets for benchmarking neurosymbolic methods, listed in ascending order of publication year. For each dataset, we provide an overview of the task, domain, modality, key characteristics, and whether the dataset is synthetic.

| Dataset | Task | Domain | Modality | Key Characteristics | Synthetic |
|---|---|---|---|---|---|
| **bAbI** Weston et al. (2015) | Language Reasoning | Natural Language | Text | Basic reasoning, generalization | Yes |
| **SNLI** Bowman et al. (2015) | Logical Reasoning | Natural Language | Text | Entailment, contradiction, neutral relationships | No |
| **CLEVR** Johnson et al. (2017) | Visual Reasoning | Computer Vision | Images & Text | Object counting, comparison, querying attributes | Yes |
| NLVR Suhr and Artzi (2017) | Visual Reasoning | Computer Vision | Images & Text | Visual reasoning, natural language understanding | Yes |
| **Sort-of-CLEVR** Santoro et al. (2017) | Relational Reasoning | Computer Vision | Images & Text | Spatial and relational reasoning | Yes |
| Visual Genome Krishna et al. (2017) | Visual Reasoning | Computer Vision | Images & Text | Object recognition, relationships, attributes | No |
| **Aristo** Clark et al. (2018) | Science Reasoning | Natural Language | Text | Natural language understanding, applying knowledge | No |
| **COG** Yang et al. (2018) | Cognitive Capabilities | Computer Vision | Images & Text | Temporal and logical reasoning | Yes |
| **MetaQA** Zhang et al. (2018) | Multi-hop Reasoning | Graph | Knowledge Graph | Multi-step reasoning, knowledge base | No |
| **SCAN** Lake and Baroni (2018) | Compositional Generalization | Command-based Language | Text | Understanding and generating novel commands | No |
| **Math Dataset** Saxton et al. (2019) | Math Reasoning | Natural Language | Text | Language understanding, symbolic reasoning | No |
| **GQA** Hudson and Manning (2019) | Visual Reasoning | Computer Vision | Images | Text & Spatial and relational reasoning | No |
| **ROAD-R** Giunchiglia et al. (2022) | Visual Reasoning | Computer Vision | Videos & (handcrafted) Logical Rules | Logical reasoning | No |

we obtained via the sweeps. The random baseline did not require hyperparameter finetuning. We also used Weights & Biases for monitoring our experiments.

Table 5: The results of a random hyperparameter search, presenting the chosen hyperparameters for different datasets and baseline models. The hyperparameters include *batch size, embedding size, learning rate, biases usage, and initialization method.* The batch size indicates the number of training subgraphs processed together before updating the model. The embedding size represents the dimensionality of the entity and relation embeddings. The learning rate controls the step size taken during model optimization. The biases denote whether bias terms are included in the model, and the initialization method refers to the technique used to initialize the model's parameters.

| Dataset | Model | Batch Size | Emb. | Learning Rate | Biases | Init. |
|---|---|---|---|---|---|---|
| syn-paths | transe | 4096 | 1531 | 7.029817939842623e-05 | False | uniform |
| syn-paths | distmult | 4096 | 158 | 0.0697979730927795 | False | uniform |
| syn-paths | complex | 4096 | 587 | 5.264944612887405e-05 | False | uniform |
| syn-tipr | transe | 2048 | 147 | 0.0008716274682049251 | True | normal |
| syn-tipr | distmult | 2048 | 168 | 0.005497983171450242 | True | normal |
| syn-tipr | complex | 2048 | 350 | 0.0015597556675205502 | True | normal |
| syn-types | transe | 2048 | 376 | 0.003017403610019781 | True | uniform |
| syn-types | distmult | 2048 | 273 | 0.0006013105272716594 | True | uniform |
| syn-types | complex | 2048 | 996 | 5.603405855158606e-05 | False | uniform |
| wd-movies | transe | 4096 | 68 | 0.000638003263107625 | False | normal |
| wd-movies | distmult | 4096 | 181 | 0.00307853821840767 | True | uniform |
| wd-movies | complex | 4096 | 102 | 0.019520125878695407 | False | uniform |
| wd-articles | transe | 32 | 888 | 6.094053758340765e-05 | True | normal |
| wd-articles | distmult | 32 | 65 | 0.03833121378755901 | False | uniform |
| wd-articles | complex | 32 | 283 | 0.002251396972378282 | False | normal |

## 7.5 Analysis of KGE Failure Modes

To help researchers diagnose model failures, the IntelliGraphs Python library includes a `verify_dataset` function that reports which logical rules a generated subgraph violates (`https://github.com/thiviyanT/IntelliGraphs`).

When we examined invalid subgraphs generated by our KGE baselines on `syn-paths`, three failure patterns appeared repeatedly:

1. **Branching paths**: The models frequently assign high probability to multiple outgoing edges from the same city. For instance, a model might predict both (Amsterdam, `train_to`, Rotterdam) and (Amsterdam, `drive_to`, Utrecht), producing a branching structure instead of a linear path.

2. **Incomplete transport coverage**: Valid paths must use all three transport modes exactly once. KGE models tend to over-represent whichever mode appears most frequently in training, often generating paths that use `train_to` twice while omitting `cycle_to` entirely.

3. **Disconnected segments**: Because each triple is scored independently, models sometimes generate edges that do not share endpoints—for example, predicting (Amsterdam, `train_to`, Rotterdam) alongside (Utrecht, `drive_to`, Groningen), leaving a gap in the path.

These failure modes all stem from the same root cause: KGE models have no mechanism to condition later predictions on earlier ones. The probability assigned to a triple depends only on the learned embeddings, not on which other triples have already been selected.

## 7.6 Semantics of IntelliGraphs

Logical rules provide a formal framework for expressing and reasoning about the semantics of a system. In this section, we discuss the logical rules we use to verify the semantics of the IntelliGraphs datasets. We express each logical rule using First-Order Logic (FOL) unless otherwise stated. We opted for First Order Logic (FOL) as the formal language to communicate logical constraints due to its ability to effectively express the necessary constraints and its widespread understanding within the machine learning community [18].

Although we provide the general FOL rules to check the semantics of graphs of *any arbitrary lengths*, we apply a size constraint (*i.e.* checking for graphs with a fixed number of triples) for the synthetic datasets. This is because the synthetic data generator produces graphs with fixed length and we defined it as part of our semantics. The size constraint can also be expressed in FOL, but we specify this constraint in *natural language* for brevity.

Traditionally, a reasoning engine is used to check logical consistencies in knowledge bases. We wrote a semantic checker in Python. This was more convenient to use within our framework as the graphs could be evaluated, without having to manually load them into a reasoning engine individually. Our semantic checker was written to closely follow the logical rules, and it is accessible through the IntelliGraph Python package.

---

18. These FOL logical constraints can also be rewritten into data specification languages, such as DataLog.

### 7.6.1 Logical Rules of syn-paths

$$\forall x, y, z : connected(x, y) \land connected(y, z) \Rightarrow connected(x, z)$$
$$\forall x, y : edge(x, y) \Rightarrow connected(x, y)$$
$$\exists x : root(x)$$
$$\forall x, y : root(x) \land root(y) \Rightarrow x = y$$
$$\forall x : root(x) \Leftrightarrow \forall y : \neg edge(y, x)$$
$$\forall x, y : connected(x, y) \Rightarrow x \neq y$$
$$\forall x : root(x) \Rightarrow \forall y : (connected(x, y) \lor x = y)$$
$$\forall x, y, z : edge(y, x) \land edge(z, x) \Rightarrow y = z$$
$$\forall x, y, z : edge(x, y) \land edge(x, z) \Rightarrow y = z$$
$$\forall x, y : edge(x, y) \Leftrightarrow cycle\_to(x, y) \lor drive\_to(x, y) \lor train\_to(x, y)$$
$$\exists x, y : cycle\_to(x, y) \land \neg \exists x', y'(cycle\_to(x', y') \land (x' \neq x \lor y' \neq y))$$
$$\exists x, y : drive\_to(x, y) \land \neg \exists x', y'(drive\_to(x', y') \land (x' \neq x \lor y' \neq y))$$
$$\exists x, y : train\_to(x, y) \land \neg \exists x', y'(train\_to(x', y') \land (x' \neq x \lor y' \neq y))$$

### 7.6.2 LOGICAL RULES OF SYN-TYPES

$$\forall x, y : spoken\_in(x, y) \Rightarrow language(x) \land country(y)$$

$$\forall x, y : could\_be\_part\_of(x, y) \Rightarrow city(x) \land country(y)$$

$$\forall x, y : same\_type\_as(x, y) \Rightarrow (language(x) \land language(y))$$
$$\lor (city(x) \land city(y)) \lor (country(x) \land country(y))$$

$$\forall x : language(x) \Rightarrow \neg country(x) \land \neg city(x)$$

$$\forall x : country(x) \Rightarrow \neg language(x) \land \neg city(x)$$

$$\forall x : city(x) \Rightarrow \neg language(x) \land \neg country(x)$$

$$\forall x, y, z : (connected(x, y) \land connected(y, z)) \Rightarrow connected(x, z)$$

$$\forall x, y : relationship(x, y) \Rightarrow (spoken\_in(x, y) \lor could\_be\_part\_of(x, y) \lor same\_type\_as(x, y))$$

$$\forall x, y : relationship(x, y) \Rightarrow x \neq y$$

### 7.6.3 LOGICAL RULES OF SYN-TIPR

$$\forall x, y : has\_role(x, y) \Rightarrow academic(x) \land role(y)$$

$$\forall x, y : has\_name(x, y) \Rightarrow academic(x) \land name(y)$$

$$\forall x, y : has\_time(x, y) \Rightarrow academic(x) \land time(y)$$

$$\forall x, y : start\_year(x, y) \Rightarrow time(x) \land year(y)$$

$$\forall x, y : end\_year(x, y) \Rightarrow time(x) \land year(y)$$

$$\forall x, y, z : end\_year(x, y) \land start\_year(x, z) \Rightarrow before(y, z)$$

$$\forall x : \neg has\_role(x, x)$$

$$\forall x : \neg has\_name(x, x)$$

$$\forall x : \neg has\_time(x, x)$$

$$\forall x : \neg start\_year(x, x)$$

$$\forall x : \neg end\_year(x, x)$$

$$\forall x : academic(x) \Rightarrow \neg role(x) \land \neg time(x) \land \neg name(x) \land \neg year(x)$$

$$\forall x : role(x) \Rightarrow \neg academic(x) \land \neg time(x) \land \neg name(x) \land \neg year(x)$$

$$\forall x : time(x) \Rightarrow \neg academic(x) \land \neg role(x) \land \neg name(x) \land \neg year(x)$$

$$\forall x : year(x) \Rightarrow \neg academic(x) \land \neg role(x) \land \neg name(x) \land \neg time(x)$$

$$\forall x : name(x) \Rightarrow \neg academic(x) \land \neg role(x) \land \neg year(x) \land \neg time(x)$$

### 7.6.4 Logical Rules of wd-movies

$$\forall x, y : connected(x, y) \Leftrightarrow has\_director(x, y) \lor has\_actor(x, y) \lor has\_genre(x, y)$$

$$\exists x : has\_director(x, \texttt{existential\_node})$$

$$\exists x : has\_actor(x, \texttt{existential\_node})$$

$$\forall x : x \neq \texttt{existential\_node} \Rightarrow connected(\texttt{existential\_node}, x)$$

$$\forall x, y : x \neq \texttt{existential\_node} \land y \neq \texttt{existential\_node} \Rightarrow \neg connected(x, y)$$

$$\forall x : \neg connected(x, \texttt{existential\_node})$$

$$\forall x, y : has\_director(x, y) \lor has\_actor(x, y) \Rightarrow person(y)$$

$$\forall x : \neg person(x) \lor \neg genre(x)$$

$$\forall x, y : has\_genre(x, y) \Rightarrow genre(y)$$

### 7.6.5 Logical Rules of wd-articles

$$\exists x : has\_author(\texttt{article\_node}, x)$$

$$\forall x, y : connected(x, y) \Leftrightarrow has\_author(x, y) \lor has\_name(x, y) \lor has\_order(x, y) \lor$$
$$cites(x, y) \lor has\_subject(x, y) \lor subclass\_of(x, y)$$

$$\forall x, y : connected(x, y) \Rightarrow \neg connected(y, x) \lor cites(y, x)$$

$$\forall x : \neg connected(x, x)$$

$$\forall x, y : has\_author(x, y) \Rightarrow x = \texttt{article\_node} \quad \land$$
$$(article(\texttt{article\_node}) \lor iri(\texttt{article\_node}))$$

$$\forall x : has\_author(\texttt{article\_node}, x) \Rightarrow author(x)$$

$$\forall x : author(x) \Leftrightarrow \exists y : has\_order(x, y) \land \exists y : has\_name(x, y)$$

$$\forall x, y : has\_order(x, y) \Rightarrow author(x) \land ordinal(y)$$

$$\forall x, y : has\_name(x, y) \Rightarrow author(x) \land (name(y) \lor iri(y))$$

$$\forall x, y, z : has\_order(x, y) \land has\_order(x, z) \Rightarrow y = z$$

$$\forall x, y, z : has\_name(x, y) \land has\_name(x, z) \Rightarrow y = z$$

$$\forall x : subject(x) \Rightarrow \neg ordinal(x) \land \neg author(x)$$

$$\forall x : iri(x) \Rightarrow \neg ordinal(x) \land \neg author(x)$$

$$\forall x : name(x) \Rightarrow \neg ordinal(x) \land \neg author(x)$$

$$\forall x : ordinal(x) \Rightarrow \neg subject(x) \land \neg iri(x) \land \neg name(x) \land \neg author(x)$$

$$\forall x : author(x) \Rightarrow \neg subject(x) \land \neg iri(x) \land \neg name(x) \land \neg ordinal(x)$$

$$\forall x, y, z : subclass\_trans(x, y) \land subclass\_trans(y, z) \Rightarrow subclass\_trans(x, z)$$

$$\forall x, y : subclass\_of(x, y) \Rightarrow subclass\_trans(x, y) \land$$
$$(iri(x) \lor subject(x)) \land (iri(y) \lor subject(y))$$

$$\forall x, y : subclass\_of(x, y) \Rightarrow \exists z : subclass\_trans(x, z) \land has\_subject(\texttt{article\_node}, z)$$

$$\forall x, y : cites(x, y) \Rightarrow article(x) \land iri(y)$$

$$\forall x, y : has\_subject(x, y) \Rightarrow article(x) \land (subject(y) \lor iri(y))$$

In addition to the aforementioned rules for `wd-articles`, our semantic checker checks the ordinal of the author's position to make sure that they are a complete list of consecutive numbers (*i.e.* `ordinal_000`, `ordinal_001`, `ordinal_002`, ..., etc.), but we leave it out of the rules for brevity.

## 7.7 Synthetic Dataset Generation

The synthetic dataset generator contains two main modules: 1) a *triple sampler* is a module that samples new triples one by one, and 2) a *triple verifier* module checks each triple for semantic validity before they are added to a subgraph. The generator builds a subgraph by sampling one triple at a time and verifying. If the triple passes the semantic check, it is added to a subgraph. To avoid duplicate triples within the same subgraph, we check if a triple already exists before adding it to a subgraph. This is done until a certain number of valid triples are sampled. For reproducibility, we use the same seed for all random data generations (`seed=42`). For each dataset, we generate *training*, *validation* and *test* sets. To avoid data leakage, we check that these graphs are unique before splitting the dataset. In this section, we briefly describe how IntelliGraphs efficiently samples valid subgraphs.

### 7.7.1 `syn-paths`

The entities are labelled after 49 Dutch cities and the relations are different modes of transport (`train_to, drive_to, cycle_to`). This dataset primarily checks whether baseline models can do structure learning.

A path graph $P_k(G)$ of a graph $G$ has vertex set $\Pi_k(G)$ and edges joining pairs of vertices that represent two paths $P_k$, the union of which forms either a path $P_{k+1}$. We denote by $\Pi_k(G)$ the set of all paths of $G$ on $k$ vertices ($k \geq 1$), and we randomly sample $n$ edges from $\Pi_k(G)$ to generate each path graph.

To generate a path graph, we begin by selecting a head (*i.e.* source node) by randomly selecting a Dutch city and then we sample relation and a tail (*i.e.* target node). For the next triple in the subgraph, we use the previous target node as the source node and then sample a relation and a target node. We can repeat the last step $k - 2$ a number of times to build a path graph with $k$ edges. We ensure that each subgraph includes all three different relations. We avoid generating cyclic path graphs.

### 7.7.2 `syn-types`

This dataset contains three types of entities (`cities`, `countries` & `languages`), 30 entities in total (10 instances of each entity type), and three relations (`same_type_as`, `could_be_part_of` & `could_be_spoken_in`). This dataset primarily checks whether baseline models can learn the types of entities correctly.

For each relation, we sample a head and a tail entity of the corresponding type. For instance, for the relation `could_be_spoken_in` we sample a language for the head of a triple and a country for the tail. Similarly, we sample other triples to be added to the same subgraph, until a certain number of valid triples have been sampled.

It is important to note that the `syn-types` dataset is not meant to be factually accurate but rather serves as a way to study the type semantics learned by machine learning models.

### 7.7.3 SYN-TIPR

This dataset contains three entity types (`names`, `roles`, `years`) and (`has_role`, `has_name`, `start_year` and `end_year`). We used a random name generator [19] to generate 50 names. For simplicity, we treat *years* as entities rather than literals. In each subgraph, there are two existential nodes: `_academic` and `_time`). The main purpose of this dataset is to check structure learning and check *basic* temporal reasoning (in this case, whether `end_year` appears after `start_year`).

The subgraphs in this dataset were modelled after the *time-indexed person role* (tipr) pattern in Semantic Web. For generating these subgraphs, we take the tipr pattern as a template and randomly sample entities with the correct entity type. For instance, the relation `has_role` would always have an academic_node in the head position of a triple and a role as a tail. Similarly, we sample triples for the other relations (`has_name`, `has_time`, `start_year`, `end_year`). Valid triples containing every relation is sampled. In total, every subgraph will contain five triples.

## 7.8 Wikidata Dataset Generation

For reproducibility, we use a specific Wikidata dump to extract the data, rather than the live version. For both datasets, we use the Wikidata HDT dump from 3 March 2021, available from the HDT website [20].

In both cases, we first extract all data that fits the template of the graph, for instance, for every movie we extract all actors, directors and genres. We then *prune* this data to ensure that every entity occurs in enough instances to allow a model to learn a representation for it. Depending on the dataset we either remove the infrequent nodes or replace them with existential nodes. We set the minimal frequency to 6 in both datasets.

To avoid the situation where certain entity nodes are only present in the validation or test data, we must make our splits carefully. Ideally, we'd like for each entity to be present in all three splits of the data, and where this is not possible, for it to be present in at least the training data.

To achieve this, we use the following algorithm: for each instance, we collect "votes" among all its entities for which of the three splits it should be part of. Simultaneously, for each entity, we collect the splits of which it is a member. The aim is to have all entities in each instance vote for the same split, and for each entity to be represented in all splits. We first alternate fixing one of the two problems: we unify the votes by choosing a random entity and setting the votes of the other entities in the instance to that vote. After all votes have been fixed, we fix the split memberships by, for each entity that is not represented in all splits, taking the most frequent split and changing the vote of one of its instances to the missing split, repeating until all splits are represented.

We alternate these steps for 50 iterations. Then, in the first step, we move any instance with conflicting votes to the training data and repeat the iteration in this fashion for another 20 steps. For both datasets, this leads to all entities being represented in the training data, and only a small number present in only the test or only the training data.

---

19. https://www.behindthename.com/random/
20. https://www.rdfhdt.org/datasets/

For both datasets, the labels are Wikipedia IRIs, but a mapping to human-readable labels is provided. In this paper, we replace IRIs with these for readability.

### 7.8.1 WD-MOVIES

We collect all entities that are labelled as "instances of" the class "film". For each we extract all entities connected by the relations "cast member", "director" and "genre" as its actors, directors and genres respectively.

We then prune the data by removing all actors, directors and genres that do not appear in at least 6 instances. We then remove any movies that are left with no actors or no directors. We allow movies with no genres. We iterate these two steps until no changes are made. Finally, we make a test, train and validation split by the process described above. The following Wikidata properties and entities were used:

| label | wikidata IRI |
|---|---|
| instance of | http://www.wikidata.org/prop/P31 |
| film | http://www.wikidata.org/entity/Q11424 |
| cast member | http://www.wikidata.org/prop/P161 |
| director | http://www.wikidata.org/prop/P57 |
| genre | http://www.wikidata.org/prop/P136 |

### 7.8.2 WD-ARTICLES

We collect all entities from Wikidata that are the object of a triple with the relation "cites".

For each article we collect the full list of authors, using the relations "author" and "author name string". The former is used to refer to authors that are represented in Wikidata as an entity, and the latter is used for authors represented only by their name as a string literal. We require at least one of the authors to be represented by an entity. If not, the article is filtered out.

Such statements are commonly annotated in Wikidata with an *ordinal*, representing the order of the author in the author list. We extract these as well. If any author does not have an ordinal or if the collection of these ordinals does not coincide exactly with the sequence $1, \ldots, n$, with $n$ the number of authors, the article is filtered out.

We then collect all articles that, as recorded in Wikidata, the current article cites. If there are no such references, the article is filtered out.

Finally, we collect the article subjects, and for each subject, every superclass and its superclass, that are an instance of "academic discipline". We do not filter based on the subjects (no subjects or superclasses are allowed).

We collect the first 100 000 such articles for the dataset `wd-articles`, and all such articles for the dataset `wd-articles-large`.

As with `wd-movies`, we prune the data to eliminate any entities that occur in fewer than 6 instances. For the authors, the article itself and the subjects, we replace these with existential nodes. These have node labels specific to the role they play in the graph: `_article`, `_author001`, and `_subject001`. Any references to infrequent entities are removed. As before, this removal process is iterated until the dataset stabilizes.

Splits are then made using the algorithm described above. In the construction of the dataset, we add authors by introducing a blank node (using label `_authorpos` and the

relation `has_author`), to which the author identity (`has_name`) and the ordinal (`has_order`) are connected. References are added by a single edge with the relation `cites` and subjects and superclasses with the relations `has_subject` and `subclass_of`.

The following Wikidata properties were used.

| label | wikidata IRI |
|---|---|
| cites | `http://www.wikidata.org/prop/P2860` |
| author | `http://www.wikidata.org/prop/P50` |
| author name string | `http://www.wikidata.org/prop/P2093` |
| main subject | `http://www.wikidata.org/prop/P921` |
| subclass of | `http://www.wikidata.org/prop/P279` |
| academic discipline | `http://www.wikidata.org/entity/Q11862829` |

## 8 Datacard

An up-to-date version of the data card can be found on `https://github.com/thiviyanT/IntelliGraphs/blob/main/Datacard.md`.

