# OpenReview forum: "IntelliGraphs: Datasets for Benchmarking Knowledge Graph Generation"
_DMLR — Accepted by DMLR_

### Review · Reviewer_V65c · 2024-11-26

**Recommendation:** 4
**Confidence:** 2

**Summary Of Contributions:**

This paper presents a new collection of five datasets for generative knowledge graph research. In each dataset, each sample is a subgraph sampled from the original KG, and the task of subgraph inference is to generate a subgraph that could pass the constraint verifier with a given set of rules. Two different kinds of evaluations for generative models are also conducted. The first one is to evaluate the NLL (with base 2) on the test dataset. The second one is to assess the percentage of novelty and the validity of the generated graph.

**Strengths:**

See above.

**Audience:**

Yes

**Claims And Evidence:**

Yes

**Datasets And Benchmarks:**

Yes

**Extended Submissions:**

NA

**Limitations:**

One possible limitation is the domain is about knowledge graphs in a transductive setting. I am curious about the inductive setting. However, it is far beyond the scope of this paper and is not part of the weakness.

**Requested Changes:**

1. Please conduct some discussion on those two types of evaluation
2. If possible, could you please conduct some experiments of simple graph generative model on the train/valid/test split?

**Strengths And Weaknesses:**

Strength:
- The subgraph inference task is novel and appears to be a very important and exciting research direction.
- Some important features are included in the dataset and the companion software, such as the n-ary predicate in syn-paths and logical constraint verifier.

Weakness:
- The major weakness is the baseline. It is pretty simple to use only the link predictor and some decomposition of the joint probability without any interdependency. Is the KGE trained on the training set of subgraphs?
- Some discussions are not well completed. As a puzzling part for me, why random model baseline has a shorter codelengths but much worse performance?

---

### Review · Reviewer_NMH8 · 2025-10-12

**Recommendation:** 4
**Confidence:** 2

**Summary Of Contributions:**

This paper introduces IntelliGraphs, a benchmark designed to evaluate knowledge graph generation models with explicit logical constraints. The authors propose a new task named subgraph inference, which aims to generate interconnected sets of triples that satisfy complex semantic rules rather than traditional link prediction. The paper includes three synthetic datasets and two real-world datasets extracted from Wikidata by formal logical constraints expressed in First-Order Logic. Baseline experiments using adapted KGE models (TransE, DistMult, ComplEx) demonstrate that traditional approaches struggle to capture complex semantics, presenting new challenges for NLP/ML researchers to explore.

**Strengths:**

The paper makes several contributions that advance knowledge graph research. It introduces a novel task named subgraph inference which can capture the interdependent patterns in KGs. The benchmark is comprehensive, combining three synthetic datasets with controlled semantic complexity and two real-world Wikidata-based datasets. In addition, this paper demonstrates strong reproducibility practices by providing open-source code, distributing the datasets through multiple channels (GitHub, Zenodo, PyPI, Conda), including detailed hyperparameters and experimental configurations, and offering comprehensive documentation with a datacard.

**Audience:**

Yes

**Broader Impact Concerns:**

No major concerns on the ethical implications of the work

**Claims And Evidence:**

The claims are supported by datasets, benchmarks, open-source code, and quantitative results.

**Datasets And Benchmarks:**

There is sufficient detail on data collection and organization, availability and maintenance, and ethical and responsible use. There are sufficient details to support reproducibility.

**Extended Submissions:**

NA

**Limitations:**

The paper has several limitations shown above: The baseline evaluation is narrow, testing only three traditional KGE models (TransE, DistMult, ComplEx). Some more sophisticated graph generation methods (e.g., GraphRNN, GraphVAE) can be evaluated. Scalability concerns on the dataset sizes (the real-world datasets contain only 60k-100k entities) and the computational overhead of checking all logical constraints for generated subgraphs, with insufficient discussion of how the approach would scale to large-scale KGs with millions of entities. In addition, while the paper successfully demonstrates that current baselines fail to generate valid subgraphs, it provides insufficient analysis of why they fail, for example, % of valid graphs have many 0s in Table 3, and more insights can be provided.

**Requested Changes:**

- Baseline Selection: More baselines can be selected and discussed, including graph models and even LLMs.
- Analysis of Failure Modes: Can you provide more analysis on the failure of models? Because of the difficulty of tasks or added constraints?
- Evaluation metrics. Can you evaluate the performance also on traditional evaluation metrics in KG resaerch?

**Strengths And Weaknesses:**

**Strengths**
- Novel and Well-Motivated Task: The subgraph inference task is novel from traditional KG tasks.
- Semantic Specification: Each dataset includes formal logical constraints in First-Order Logic, with accompanying constraint verifier functions.
- Comprehensive Benchmark: The combination of synthetic datasets (with controlled complexity) and real-world datasets (with natural complexity) provides a well-rounded evaluation framework.
- Strong Reproducibility: The authors provide open-source code, Python packages (via PyPI and Conda), datasets on Zenodo, detailed hyperparameters, and documentation.

**Weakness**
- Limited Baseline Diversity: Only three traditional KGE models are adapted as baselines.
- Evaluation metrics. This paper proposes some new evaluation metrics for this new task, but it may be helpful to conduct the experiment with existing link prediction related metrics.
- Insufficient Analysis of Failure Modes: While the paper shows that KGE baselines fail, there is limited analysis of why they fail or what specific semantic aspects are most challenging.
- Scalability Concerns: The real-world datasets are relatively small.

---

### Review · Reviewer_ZPKH · 2025-11-06

**Recommendation:** 4
**Confidence:** 2

**Summary Of Contributions:**

The paper recognizes that existing link prediction datasets like FB15k-237 and WN18RR, are not based on well-defined logical constraints. Which makes testing whether the model learned the semantics of a dataset impossible.

That is why the authors propose a new task: Subgraph inference, where the goal is to generate, from a set of training examples that follow certain logical rules, novel subgraphs that are coherent to the rules in the training data. Link predictors would not be good at this task since they predict links independently from eachother and this task involves predicting a set of interdependent links.

IntelliGraphs is a new benchmark for the subgraph inference problem. Containing 5 datasets, three synthetic and two derived from Wikidata.

First experiments on this benchmark show that KGE models have a limited understanding of semantics after training. The probability of sampling a valid graph was consistently low, or even 0, emphasizing the difficulty of the task and the need for the benchmark.

**Strengths:**

* The need for the benchmark is clear. How the task differs from link prediction and why existing benchmarks do not suffice.
* The 5 proposed datasets are appropriate for the task. Simple logical constrains, yet difficult to capture.
* Good documentation. Not just the datasets but also code to reproduce is available both on Github and as Python package.
* Results show that KGE models do not capture semantics (which was to be expected)

**Audience:**

Yes

**Broader Impact Concerns:**

No impact concerns. The datasets are synthetic or sourced from publicly available Wikidata, mitigating privacy or bias risks. The benchmark encourages development of reasoning-aware graph models, which has positive implications for scientific and industrial AI.

**Claims And Evidence:**

Claims:
1) Subgraph inference is a new task of generating subgraphs coherent to First-Order Logic semantics.
* Yes. Well argued how it differs from link prediction and why the task is important.
2) IntelliGraphs datasets provide a robust benchmark for the subgraph inference task.
* Yes. Detailed documentation, open release and diverse datasets well defined for this task.
3) Existing KGE models fail to capture semantics.
* Yes. Convincingly supported both theoretical and with quantitative results.

**Datasets And Benchmarks:**

Yes. Not just the datasets but also code to reproduce is available both on Github and as Python package.

**Extended Submissions:**

-

**Limitations:**

* The models tested are deliberately weak on semantics; hence, the benchmark’s full difficulty isn’t demonstrated against more capable architectures.
* Nice to see an evaluation of the code-length, although conclusions are not that clear to me. Do outcomes suggest that the random baseline model is better at compressing the graph?
*  Paper would benefit from some more visuals/examples. In explaining the 5 datasets, but also in how KGE models make invalid graphs by nature.

**Requested Changes:**

Changes that would strengthen:
* The paper only evaluates KGE models, which are already expected to be bad at the subgraph inference task. Would it not be better to also test more expressive baselines? Maybe graph generative models, or variational graph autoencoders? (unsure if these models can be expected to learn semantics).
*  Paper would benefit from some more examples. It would help having a visual, like figure 1, for each dataset. The visual of syn-paths allowed me to understand the problem in a glance. For the other datasets I had to dive deeper in the appendix.
* Maybe also an example of how KGE models make invalid graphs by independently generating edges with high individual likelihoods.

**Strengths And Weaknesses:**

Im using the strenghts and limitation boxes below to not repeat myself.

---

### Review · Reviewer_EbxA · 2025-11-19

**Recommendation:** 4
**Confidence:** 2

**Summary Of Contributions:**

This paper studied the problem of knowledge graph embedding by highlighting the impact of semantics in subgraphs. This paper introduced the subgraph inference task to explore the logical consistency among subgraphs. A set of new datasets (IntelliGraphs) was created for complex semantic understanding in knowledge graphs. Evaluation results also showed that the existing KGE approaches might not well capture such complex semantics.

**Strengths:**

(S1) This work highlights the semantic understanding in knowledge graphs by introducing the subgraph inference task.

(S2) The IntelliGraphs benchmark is proposed to advance the research of knowledge graph embedding with complex semantic understanding.

(S3) Experimental evaluation demonstrates that KGE-based baselines cannot capture complex semantics. It thus motivates the development of advanced KGE approaches by understanding fine-grained semantics in knowledge graphs.

**Audience:**

Yes

**Claims And Evidence:**

The claims in semantic understanding are well supported in the experiments.

**Datasets And Benchmarks:**

The authors provide sufficient detail on data collection and organization, availability and maintenance, and ethical and responsible use.

**Extended Submissions:**

No

**Limitations:**

Please see above

**Requested Changes:**

It would be more convincing to clarify the data generator process in IntelliGraphs below.

(W1) The mechanisms of the data generator in IntelliGraphs are confusing. The data generator randomly samples subgraphs from a probability distribution over all possible valid subgraphs. Will it be computationally inefficient to sample logically consistent graphs? Besides, why can this random sampling strategy discourage the likelihood of consistent patterns between the same entities? How will it ensure that they rely on semantic understanding rather than memorized cues?

(W2) The running efficiency of IntelliGraphs for the data generation can be discussed

**Strengths And Weaknesses:**

Strengths:

(S1) This work highlights the semantic understanding in knowledge graphs by introducing the subgraph inference task.

(S2) The IntelliGraphs benchmark is proposed to advance the research of knowledge graph embedding with complex semantic understanding.

(S3) Experimental evaluation demonstrates that KGE-based baselines cannot capture complex semantics. It thus motivates the development of advanced KGE approaches by understanding fine-grained semantics in knowledge graphs.

Weaknesses:

(W1) The mechanisms of the data generator in IntelliGraphs are confusing. The data generator randomly samples subgraphs from a probability distribution over all possible valid subgraphs. Will it be computationally inefficient to sample logically consistent graphs? Besides, why can this random sampling strategy discourage the likelihood of consistent patterns between the same entities? How will it ensure that they rely on semantic understanding rather than memorized cues?

(W2) The running efficiency of IntelliGraphs for the data generation can be discussed